# Interfacial and Filler Size Effects on Mechanical/Thermal/Electrical Properties of CNTs-Reinforced Nanocomposites

**DOI:** 10.3390/polym16060808

**Published:** 2024-03-14

**Authors:** Jie Wang, Xinzhu Duan, Liangfei Gong, Shuyan Nie

**Affiliations:** 1College of Aerospace Engineering, Chongqing University, Chongqing 400044, China; jiewang_cae@cqu.edu.cn; 2State Key Laboratory of Mountain Bridge and Tunnel Engineering, Chongqing Jiaotong University, Chongqing 400074, China; xinzhuduan2024@163.com

**Keywords:** mechanical/thermal/electrical properties, interfacial effect, filler size effect, modified EMT model

## Abstract

The mechanical/thermal/electrical properties on-demand design of CNTs-reinforced nanocomposites is a key scientific issue that limits the development of new-generation smart nanomaterials, and the establishment of a corresponding unified theoretical prediction model for the mechanical/thermal/electrical properties is the foundation of nanocomposites. Based on the equivalent medium theory (EMT) obtained by Maxwell far-field matching, a unified mechanical/thermal/electrical modified EMT model is established by introducing Young’s modulus, thermal conductivity, and electrical conductivity to the thin filler–matrix’s interlayer. According to literature, the proposed model was employed to theoretically calculate the variations in the overall Young’s modulus, thermal conductivity, and electrical conductivity of CNTs-reinforced nanocomposites with respect to the volume concentration of CNT fillers. Then, the applicability of the proposed theoretical model was validated in comparison with the experimental measurements. Numerical calculations showed that the interface is a key factor affecting the mechanical/thermal/electrical properties of CNTs-reinforced nanocomposites, and strengthening the interfacial effect is an effective way to enhance the overall properties of nanocomposites. In addition, the aspect ratio of CNT fillers also significantly affects the material properties of the CNT fillers interface phase and the CNTs-reinforced nanocomposites. By fitting the experimental data, the calculation expressions of the aspect ratios of CNT fillers on the Young’s modulus, thermal conductivity, and electrical conductivity of the CNT fillers interfacial phase are quantitatively given, respectively.

## 1. Introduction

Carbon nanotubes (CNTs) have attracted extensive attention from researchers due to their various excellent properties, such as high modulus, high strength [1], high electrical conductivity [2], and high thermal conductivity [3], making them ideal reinforcement for nanocomposites [4,5]. The high aspect ratio and high specific surface area of CNTs can maximize the excellent micro-nano properties of CNT fillers on the macro scale. We can take full advantage of the synergistic effect of composites and nanotechnology to fabricate CNT nanocomposites that are lighter in mass, higher in modulus, and have functional features, such as electromagnetic shielding, heat insulation, and mechanical–electric coupling [6].

For the mechanical properties of CNTs-reinforced nanocomposites, the interlayer, as an important medium between the CNT fillers and the matrix, can not only transfer the load, but its mechanical properties also affect the deformation mechanism of the matrix and the crack extension mode and path [7,8]. Cha et al. [9] experimentally fabricated CNTs–epoxy nanocomposites and performed the tensile tests and single-edge notch bending (SENB) tests at various weight fractions of CNT fillers. They found that the Young’s modulus of the CNTs–epoxy nanocomposites was too low, compared with theoretical values, calculated by using the well-established Halpin–Tsai model [10,11]. Alishahi et al. [12] compared the experimentally measured Young’s modulus of CNTs–epoxy nanocomposites with the predicted values of the modified Halpin–Tsai [13], the Mori–Tanaka equation [14], and the Hashin and Shtrikman model [15], respectively, and found that the modified Halpin–Tsai equation has good agreement when the volume concentration of CNT fillers is low. However, when the volume concentration increases, the present theoretical model still overestimates the overall estimated Young’s modulus of the nanocomposites. It is believed that the CNT matrix interlayer, which is mainly due to the increase of CNT fillers, plays a dominant role and cannot be neglected in theoretical calculations [16]. Therefore, the interfacial bonding strength and mechanical properties between the fillers and the matrix have a crucial influence on the mechanical properties of their nanocomposites. Giovannelli et al. [17], Dehrooyeh et al. [18], Hsieh et al. [19], Mei et al. [20], Zarasvand et al. [21], and Wang et al. [22] independently fabricated CNTs-filled nanocomposites with different geometries, and there existed significant differences between the overall Young’s moduli obtained from their experimental measurements. This is mainly because the CNT fillers with large aspect ratios are not easily dispersed in the matrix, and entanglement and bending or even agglomeration can significantly reduce their reinforcement qualities [16]. Therefore, does the aspect ratio of CNT fillers affect the mechanical properties of their filler–matrix interface? This paper will discuss the issue further.

For the thermal properties of nanocomposites, the Kapitza thermal resistance of the interface is often considered to be key to the thermal conductivity of CNTs-polymer nanocomposites [23,24]. Nan et al. [25] developed a generalized equivalent medium approximation (essentially identical to the Mori–Tanaka equation) for the effective thermal conductivity of ellipsoidal particle composites. They achieved the characterization of the interfacial Kapitza thermal resistance effect by introducing a zero-thickness interfacial layer with predetermined boundary thermal resistance. As remarked by Nan et al. [26] and others [23,27], the marked disparity between the predictions and the experimental observation may be due to the interfacial contact resistance between the CNTs and the matrix. The presence of the CNT–matrix interfacial thermal resistance (also called Kapitza thermal resistance) may result in energy loss and subsequently lead to a degradation in the effective thermal conductivity. Based on this concept, Sheng et al. [28] developed a multiscale approach within the framework of EMT to predict the effects of CNT agglomeration as well as the CNT interfacial thermal resistance on the overall thermal conductivity of the nanocomposites over a wide range of CNT filler loading. This model suggests that with the differences in the densities of phonon states between CNT fillers and matrix phase materials in nanocomposites, phonons are easily scattered at the CNT–matrix interface, and thermal transport is then greatly impeded [29]. The intrinsic thermal conductivity of CNTs, as the most important filler and reinforcement in nanocomposites, is the most critical performance parameter. However, it was found that the aspect ratio of CNTs has a significant role in influencing the overall thermal properties of their nanocomposites [30,31,32,33,34,35,36].

For the electrical properties of nanocomposites, adding highly conductive CNTs to the insulating polymer matrix can dramatically improve the electrical conductivity of polymer nanocomposites. The CNT fillers constructed a conductive network within the matrix, which formed a conductive pathway, and the conductive behavior of polymer-conducting nanocomposites is generally consistent with the conventional percolation theory [37]. Therefore, the highly conductive CNT filler with a high aspect ratio is conducive to the construction of the percolation conductive network, which means that less filler is needed to complete the construction of the conductive network and reach the conductive percolation. However, experimental studies [22,34,35,36,38,39,40,41] have found that CNTs with high aspect ratios also lead to weakened enhancement. Wang et al. [42] developed a continuum model that possesses several desirable features of the electrical conduction process in CNTs-based nanocomposites. The interfacial resistance was characterized by the introduction of a diminishing layer of interface with an interfacial conductivity to build a “thinly coated” CNT. They demonstrated that this model could successfully capture the quantitative behaviors of two sets of experimental data, one for multi-walled CNTs and another for single-walled CNTs-polyimide nanocomposites.

To obtain on-demand CNTs-polymer nanocomposites with a combination of excellent properties, such as high modulus, high strength, high thermal conductivity, and high electrical conductivity, it is necessary to comprehensively consider the effects of various factors, such as CNTs–polymer interfacial connection, CNT filler aspect ratio, and dispersion state. So far, there is a lack of a unified theoretical model to investigate the basic rules governing the effects of the above factors on the mechanical/thermal/electrical properties of CNTs-reinforced nanocomposites. In this paper, based on the effective medium theory (EMT) obtained via Maxwell’s far-field equivalence principle [43], the theoretical prediction model of the overall Young’s modulus, thermal conductivity, and electrical conductivity of the CNTs-reinforced nanocomposites is derived by introducing the CNT filler aspect ratio-dependent interfacial modulus and the thermal and electrical conductivities. The expressions for the relationship between the CNT aspect ratio and the interfacial modulus and the thermal, and electrical conductivities are given. Finally, to verify the applicability of the proposed model, the theoretically calculated values are compared with the experimentally measured values.

## 2. Theory

### 2.1. Coated CNT Fillers

To make the theoretical model available for better applicability and to obtain a simple analytical form, the shape of the CNT filler is idealized as a three-dimensional solid cylinder. Figure 1 shows a schematic diagram of 3D cylindrical CNT fillers with length *l* and diameter *λ*. *α* = *l*/*λ* is the aspect ratio of CNT fillers.

A cell element consists of a two-phase material, a filler of CNTs (with the upper or lower label f representing the relevant variable), and a thin interfacial layer surrounding the surface of the CNTs (with the upper or lower label int representing the relevant variable).

The modulus tensors **L**^int^ and **L**^f^ (which can be replaced by the material elastic modulus **E**, thermal conductivity tensor **κ**, and electrical conductivity tensor **σ** in different research areas) and the corresponding volume concentrations *V*_int_ and *V*_f_ are introduced for the interfacial and reinforced phases, respectively. The interfacial layer fits very closely with the CNT fillers, whose thickness is thin enough to be an order of magnitude smaller than the diameter of the CNTs [28,42]. Owing to the same aspect ratio of CNT fillers and interfacial layers, the equivalent modulus tensor of CNTs fully wrapped by the interfacial layer can be obtained using the Mori–Tanaka method as shown [23,24]:(1)$$L^{c-f}=L^{\mathrm{int}}\left[1+\frac{\left(1-V_{\mathrm{int}}\right)\left(L^{f}-L^{\mathrm{int}}\right)}{V_{\mathrm{int}}S^{f}\left(L^{f}-L^{\mathrm{int}}\right)+L^{\mathrm{int}}}\right]$$
where **L**^c−f^ is the equivalent modulus tensor of coated CNT fillers. $L_{1}^{f}=L_{2}^{f}$ and $L_{1}^{c-f}=L_{2}^{c-f}$ are the components of the radial direction of the moduli tensor for CNTs and coated CNT fillers, respectively, while $L_{3}^{f}$ and $L_{3}^{c-f}$ are the axial direction components of the CNTs and coated CNT filler modulus tensor, respectively. Taking direction-3 as the symmetric axis of the cylinder, the Eshelby tensor **S^f^** of CNT fillers is related only to the aspect ratio *α* of CNTs, and its computational expression is as follows [23,24]:(2)$$\begin{array}{l} S_{11}^{f}=S_{22}^{f}=\frac{\alpha }{2{\left(\alpha^{2}-1\right)}^{\frac{3}{2}}}\left[\alpha {\left(\alpha^{2}-1\right)}^{\frac{1}{2}}-\cosh^{-1}\alpha \right]\\ S_{33}^{f}=1-2S_{11}^{f}\\ S_{11}^{m}=S_{22}^{m}=S_{33}^{m}=S_{0}^{m}=\frac{1}{3} \end{array}$$

The volume concentration of the thin interfacial layer relative to the CNT fillers is
(3)$$V_{\mathrm{int}}=1-\left[\left(\alpha \lambda \right)\cdot {\left(\frac{\lambda }{2}\right)}^{2}\right]/\left[\left(\alpha \lambda +2\delta \right)\cdot {\left(\frac{\lambda }{2}+2\delta \right)}^{2}\right]$$

### 2.2. CNTs-Reinforced Nanocomposites

After randomly distributing the coated CNT fillers in the polymer matrix, a tiny unit containing a two-phase material consisting of the matrix (with the upper or lower label m representing the relevant variable) and the coated CNT fillers (with the upper or lower label c-f representing the relevant variable) is considered. The modulus tensors of the matrix and composite-reinforced phases are **L**^m^ and **L**^c−f^ and the corresponding volume concentrations *c*_m_ and *c*_c−f_ = *c*_f_/(1 − *V*_int_), respectively. According to Maxwell’s far-field equivalence condition, the volume average of the scattered fields **T**^m^ and **T**^c−f^ of all component phases in the composite at a distance is equal to the scattered field **T**^e^ of the equivalent medium.
(4)$$T^{e}=c_{m}T^{m}+c_{c-f}T^{c-f}$$
where
(5)$$\left\{\begin{array}{l} T^{m}={\left[{\left(L^{m}-L^{r}\right)}^{-1}+S^{m}{\left(L^{r}\right)}^{-1}\right]}^{-1}\\ T^{c-f}={\left[{\left(L^{c-f}-L^{r}\right)}^{-1}+S^{c-f}{\left(L^{r}\right)}^{-1}\right]}^{-1} \end{array}\right.$$

Under the assumption of the same aspect ratio of CNT fillers and interfacial layers, the depolarization tensor (Eshelby tensor) of coated CNT fillers **S**^c−f^ = **S^f^**. Bringing in the homogenization equivalence condition, the reference medium modulus tensor must be equal to the equivalent medium modulus tensor, i.e., **L**^r^ = **L**^e^. Applying a spatial orientation-averaging calculation to the equivalent modulus tensor **L**^e^ of the CNTs-reinforced nanocomposites yields:(6)$$\frac{c_{m}\left(L^{m}-L^{e}\right)}{L^{e}+S_{0}^{m}\left(L^{m}-L^{e}\right)}+\frac{c_{f}}{3V_{f}}\left[\frac{2\left(L_{1}^{c-f}-L^{e}\right)}{L^{e}+S_{11}^{f}\left(L_{1}^{c-f}-L^{e}\right)}+\frac{\left(L_{3}^{c-f}-L^{e}\right)}{L^{e}+S_{33}^{f}\left(L_{3}^{c-f}-L^{e}\right)}\right]=0$$

Combining Equations (1) and (6) to introduce interfacial effects, we then establish theoretical prediction models for the equivalent Young’s modulus (*L* replaced by *E*), thermal conductivity (*L* replaced by *κ*), and electrical conductivity (*L* replaced by *σ*) of CNTs-reinforced nanocomposites.

## 3. Results and Discussion

By introducing interfacial mechanical/thermal/electrical property formulas into the effective medium theoretical framework, we developed a prediction model for the overall elastic modulus, electrical conductivity, and thermal conductivity of composites considering interfacial effects. In the following, we will employ the theoretical model to calculate the mechanical, electrical, and thermal properties of CNTs-reinforced nanocomposites, respectively, and compare them with the experimental results to verify the model’s applicability.

### 3.1. Young’s Modulus and Elastic Properties

Firstly, referring to the experimental results of Javadinejad et al. [44], the average length and diameter of untreated CNTs were 15 μm and 7.5 nm, respectively, with an aspect ratio of 2000, axial Young’s modulus of 1 TPa, and radial Young’s modulus of 30 GPa. The Young’s modulus of a pure epoxy matrix was 2.6 GPa. The literature [45,46,47] and our previous work [28,48] indicated that the thickness of the interlayer is an order of magnitude smaller than the diameter of the CNTs. Considering the effect of interfacial defects in the contact region between the CNT filler and epoxy matrix on Young’s modulus of CNT–epoxy nanocomposites, the modified EMT model was used to calculate the Young’s modulus of CNT–epoxy composites. The variation curves of Young’s modulus vs. CNT volume concentration obtained from theoretical calculations were plotted as compared to the experimental observations in Figure 2a. For the reader’s convenience, we listed the values of the model parameters used in the calculations in Table 1. In order to further determine the correlation mechanism between the CNT–epoxy interfacial modulus and the filler’s aspect ratio, we performed similar theoretical calculations and a comparative analysis of the experimental data of Giovannelli et al. [17], Cha et al. [9], Alishahi et al. [12], Dehrooyeh et al. [18], Hsieh et al. [19], Mei et al. [20], Zarasvand et al. [21], and Wang et al. [22]. The comparison of the obtained theoretical calculations with the experimental measurements is plotted in Figure 2b, and Figure 2c presents the variation rule of the interfacial modulus with the aspect ratio of the CNTs.

The results show that the CNT–epoxy interfacial effect is the main factor affecting the Young’s modulus of CNT–epoxy nanocomposites, and the blue solid line in Figure 2a, which considers the interfacial effect (EMT w/Int. Eff.), is closer to the experimental results than the black dashed line without the interfacial effect (EMT w/o Int. Eff.). In order to illustrate the accuracy of the calculation of the modified EMT theory after considering the interfacial effects, the same procedure was used to calculate and analyze the results of five groups of experiments (Giovannelli et al. [17], Dehrooyeh et al. [18], Hsieh et al. [19], Zarasvand et al. [21], Wang et al. [22]) that were independent of each other, and each group of CNT fillers was randomly and uniformly embedded into the epoxy. The calculated results are shown in Figure 2b. The calculated results considering interfacial effects matched the experimental results of the Young’s modulus of the nanocomposites, and the overall Young’s modulus of the low-aspect ratio CNT–epoxy nanocomposites was higher. This is due to the more uniform dispersion and perfect adhesion of short CNTs in the epoxy resin, which play an enhanced role in the mechanical properties of the epoxy matrix. In order to clearly understand the effect of the CNT size effect on the mechanical properties of the composites, theoretical calculations of Young’s modulus results for 10 independent sets of CNT–epoxy nanocomposites were performed using the same procedure. We listed in Table 2 the aspect ratio, the mechanical properties of the interfacial layer, and the Young’s modulus of the nanocomposites for each group of CNTs-reinforced nanocomposites (*c*_f_ = 0.005).

The *α*-dependent equivalent mechanical properties of the interfacial layer of CNTs-reinforced nanocomposite are obtained, as shown in Figure 2c. The quantitative relationship between the equivalent mechanical properties of the CNT–epoxy interfacial layer and the CNT filler aspect ratio approximately satisfies an exponential rule, i.e.,:(7)$$E^{\mathrm{int}}=3.683e^{5}\alpha^{-1.158}+33.41$$

It illustrates that the increase in CNT aspect ratio leads to a decrease in the equivalent mechanical properties of the CNT–epoxy interfacial layer, eventually resulting in the decrease of the overall Young’s modulus of the nanocomposites. The strong CNT–epoxy interfacial effect can transfer the load from the epoxy matrix to the CNT fillers, and the increase in the CNT aspect ratio deteriorates the interfacial transfer effect.

### 3.2. Thermal Conductivity and Thermal Properties

For the thermal properties of CNTs-reinforced nanocomposites, we performed theoretical calculations compared to the experimental results of Yu et al. [36] and analyzed the agreement between the experimentally measured and calculated values. The CNT fillers had an average length of 5 μm, a diameter of 20 nm, an aspect ratio of 250, a radial thermal conductivity of 60 (W/mK), and an axial thermal conductivity of 3000 (W/mK). The thermal conductivity of the epoxy matrix was 0.223 (W/mK). Considering the interfacial thermal resistance (Kapiza resistance) effect of CNT–epoxy on the thermal conductivity of CNT–epoxy nanocomposites, the modified effective medium theory was used to calculate the value of thermal conductivity of CNT–epoxy nanocomposites, and the calculated and experimental results were plotted in Figure 3a. For the reader’s convenience, we listed the values of the model parameters used in the calculations in Table 3.

As shown in Figure 3a, the EMT model with consideration of the thermal resistance at the CNT–epoxy interface can accurately predict the thermal conductivity of CNTs-reinforced nanocomposites as the CNT filler volume concentration increases. In order to illustrate the accuracy of the calculation of the modified EMT model after considering the interfacial thermal resistance (Kapiza resistance), the same procedure was applied to calculate and analyze the results of five groups of experiments (Chen et al. [34,35], Yu et al. [36], and Shi et al. [32]) that were independent of each other, and each group of CNT–fillers was randomly and uniformly embedded into the epoxy. The calculated results are shown in Figure 3b. Results showed that the overall thermal conductivity of the low-aspect ratio CNT–epoxy nanocomposites was higher. This is due to the fact that a lower aspect ratio tends to boost the establishment of the CNT thermal pathways and hence increase the overall thermal conductivity [30]. 

In order to clearly understand the effect of the CNT size effect on the thermal properties of the composites, theoretical calculations of thermal conductivity for eight independent sets of CNTs–epoxy nanocomposites were performed using the same procedure. We listed the aspect ratio, the thermal properties of the interfacial layer, and the thermal conductivity of the nanocomposites for each group of CNTs-reinforced nanocomposites (*c*_f_ = 0.08) in Table 4.

The *α*-dependent equivalent thermal properties of the CNT fillers interfacial layer are shown in Figure 3c. The relationship between the equivalent thermal properties of the CNTs–matrix interfacial layer and the filler’s aspect ratio approximately satisfies the exponential rule, i.e.,:(8)$$\kappa^{\mathrm{int}}=545.4\alpha^{-0.545}-12.03$$

It illustrates that the increase in the aspect ratio of CNT fillers leads to a decrease in the equivalent thermal properties of the CNTs–epoxy interfacial layer, eventually resulting in the decrease of the overall thermal conductivity of the nanocomposites. The strong CNTs–epoxy interfacial effect enables a more efficient transfer of the heat from the matrix to the CNT fillers. An increase in the aspect ratio of CNTs can enhance the interfacial thermal resistance within the nanocomposites.

### 3.3. Electrical Conductivity and Electrical Properties

For the electrical properties of CNTs-reinforced nanocomposites, we performed theoretical calculations compared to the experimental results of Dai et al. [38] and analyzed the agreement between the experimentally measured and calculated values. The CNT fillers had an average length of 0.6 μm, a diameter of 1.5 nm, an aspect ratio of 400, a radial electrical conductivity of 5 (S/m), and an axial electrical conductivity of 5000 (S/m). The electrical conductivity of the epoxy matrix was 2.11 × 10^−11^ (S/m). Considering the effect of CNT–epoxy interfacial electrical resistance on the electrical conductivity of CNT–epoxy nanocomposites, the modified EMT model was adopted to calculate the electrical conductivity of CNT–epoxy composites, and the calculated and experimental results were plotted in Figure 4a. For the reader’s convenience, we listed the values of the model parameters used in the calculations in Table 5.

As shown in Figure 4a, the EMT model with consideration of the electrical resistance at the CNT–epoxy interface can accurately predict the electrical conductivity of CNTs-reinforced nanocomposites as the CNT filler volume concentration increases. In order to illustrate the accuracy of the calculation of the modified EMT model after considering the interfacial electrical resistance, the same procedure as in the previous section was used to calculate and analyze the results of four groups of experiments (Yin et al. [41], Chen et al. [34,35], Wang et al. [22], Yu et al. [36]) that were independent of each other, and each group of CNT fillers was randomly and uniformly embedded into the epoxy. The calculated results, shown in Figure 4b, revealed that the CNT fillers can greatly improve the electrical conductivity of the insulating epoxy matrix. The overall electrical conductivity of the low-aspect ratio CNT–epoxy nanocomposites was higher. This was due to the more uniform dispersion and perfect adhesion of the low-aspect ratio CNTs in the epoxy matrix, which play an enhanced role in the electrical properties of the insulating polymers. 

To understand the effect of the CNT size effect on the electrical properties of the CNTs-reinforced nanocomposites, theoretical calculations of electrical conductivity results for seven independent sets of CNT–epoxy nanocomposites were performed using the same procedure. We listed in Table 6 the aspect ratio, the electrical properties of the interfacial layer, and the electrical conductivity of the nanocomposites for each group of CNT-reinforced nanocomposites (*c*_f_ = 0.05).

The *α*-dependent equivalent electrical properties of the CNT fillers interfacial layer are obtained in Figure 4c. The relationship between the equivalent electrical properties of the CNT–matrix interfacial layer and the filler’s aspect ratio approximately satisfies the exponential rule, i.e.,:(9)$$\sigma^{\mathrm{int}}=29.51\alpha^{-0.837}-0.012$$

It illustrates that the increase in the aspect ratio of CNT fillers leads to a decrease in the equivalent electrical l properties of the CNTs–matrix interfacial layer, eventually resulting in the decrease of the overall electrical conductivity of the nanocomposites. The strong CNT–matrix interfacial effect enables a more efficient transfer of the electrical flux from the matrix to the CNT fillers. An increase in the CNT aspect ratio enhances the interfacial thermal resistance within the nanocomposites.

## 4. Conclusions

Based on the EMT theoretical model derived by Maxwell far-field matching, a unified force–thermal–electrical-modified EMT model is established by introducing Young’s modulus, thermal conductivity, and electrical conductivity of the fillers–matrix’s interface via the assumption of thin interfacial layers. The theoretical model can be employed to theoretically predict the overall Young’s modulus, thermal conductivity, and electrical conductivity of nanocomposites reinforced with different volume concentrations of CNT fillers by the known geometry parameters (aspect ratio) and material property parameters (Young’s modulus, thermal conductivity, and electrical conductivity) of CNT fillers and the same material property parameters of the polymer matrix. Through comparative analysis and validation with a large number of experimental results, the applicability of the model is confirmed, and the correlation relationships between the interfacial material property parameters (Young’s modulus, thermal conductivity, and electrical conductivity) and the CNT filler geometrical parameter (aspect ratio) are given, along with the computational expressions. The interaction between CNTs and the polymer matrix in the theoretical calculations is van der Waals forces, while other interfacial interactions, such as covalent and electrostatic, are not considered. The covalent and electrostatic interactions may cause an enhancement of the interfacial property, i.e., the surfactant treatment process of nanocomposites [16]. Through numerical calculations, it is found that CNT fillers with low aspect ratios are more favorable to the contact between CNTs and the polymer matrix, which greatly weakens the resistances of stress transfer, heat flow transfer, and current transfer between the reinforced phase and the matrix phase and thus significantly enhances the overall mechanical/thermal/electrical properties of CNTs-reinforced nanocomposites. It is worth noting that our theoretical model is more suitable for the case of low-volume concentration fillers. When the volume concentration of CNTs reaches an upper limit, the mechanical properties of CNTs-reinforced nanocomposites show decays due to the fillers agglomeration effect [16], while the thermal and electrical properties show enhancements, due to the fillers channeling effect [24,42]. These phenomena are not captured in our model.

## Figures and Tables

**Figure 1 polymers-16-00808-f001:**
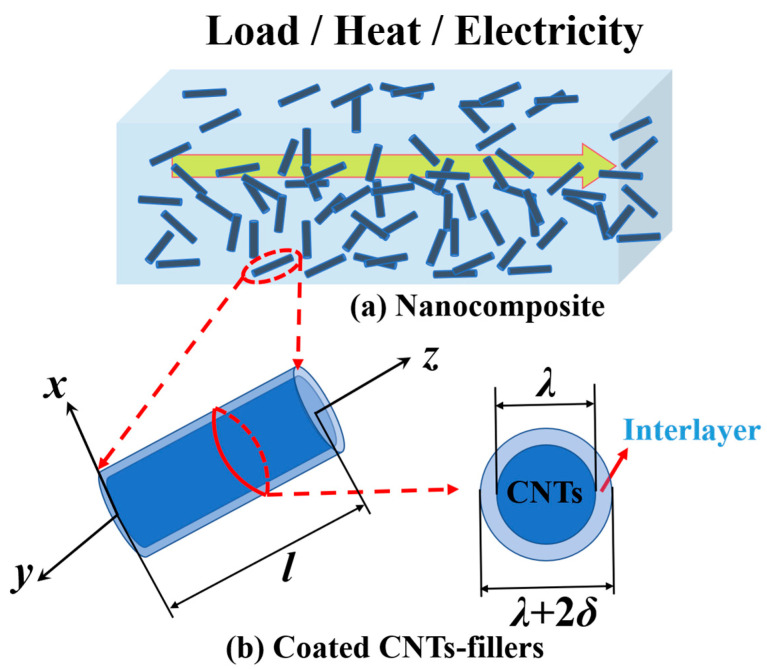
The schematics of the CNTs-reinforced nanocomposite: (**a**) the CNTs-polymer nanocomposite with randomly distributed coated CNT nanofillers; (**b**) a CNT nanofiller with the thin interlayer. *l* and λ are the length and diameter of the CNT nanofiller, *α* = *l*/*λ* is the aspect ratio, and *δ* is the thickness of the thin interlayer.

**Figure 2 polymers-16-00808-f002:**
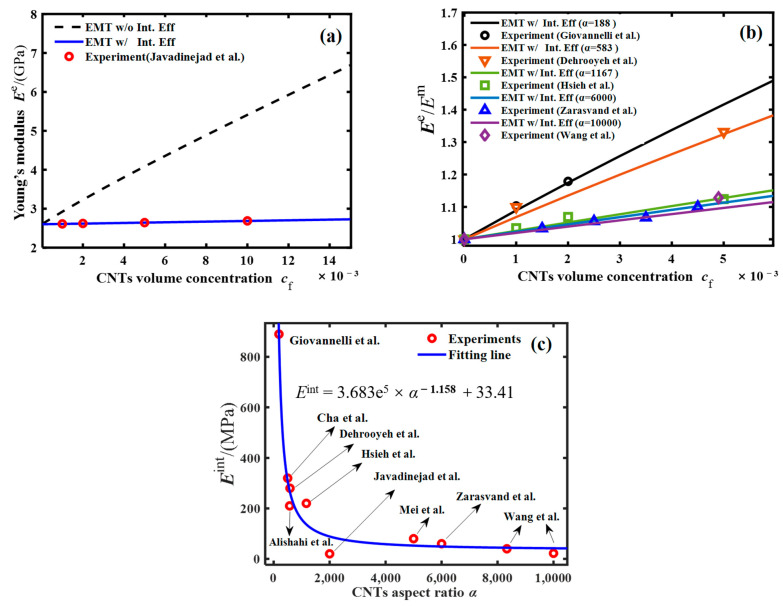
The Young’s modulus of the nanocomposite vs. the volume concentration of the CNT nanofiller: (**a**) comparison between the theoretical results and experimental measurements in literature [9,12,17,18,19,20,21,22,44], (**b**) Young’s modulus vs. the volume concentration of the CNT nanofiller; and (**c**) the interfacial modulus of CNT fillers and epoxy matrix vs. the aspect ratio of the CNT nanofiller.

**Figure 3 polymers-16-00808-f003:**
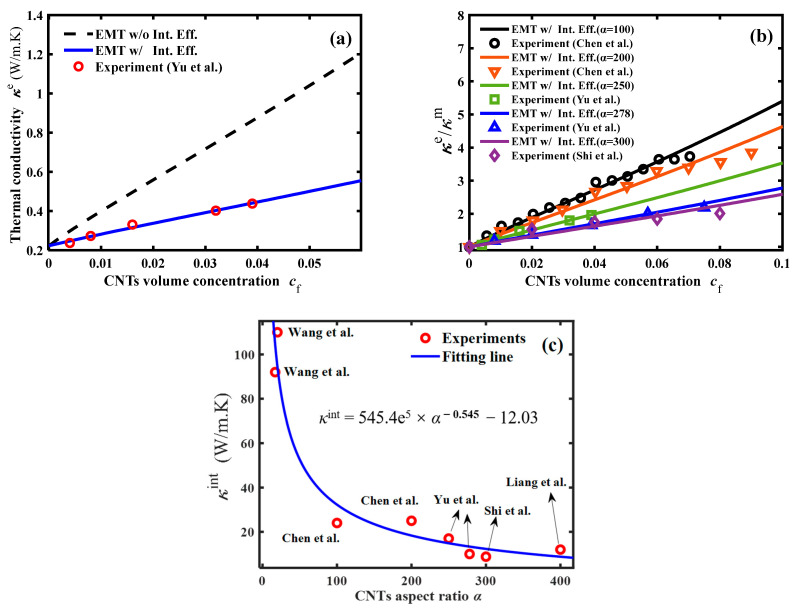
The thermal conductivity of the nanocomposite vs. the volume concentration of the CNT nanofiller: (**a**) comparison between the theoretical results and experimental measurements in literature [31,32,33,34,35,36]; (**b**) the thermal conductivity vs. the volume concentration of the CNT nanofiller; and (**c**) the interfacial thermal conductivity of CNTs and epoxy vs. the aspect ratio of CNT nanofiller.

**Figure 4 polymers-16-00808-f004:**
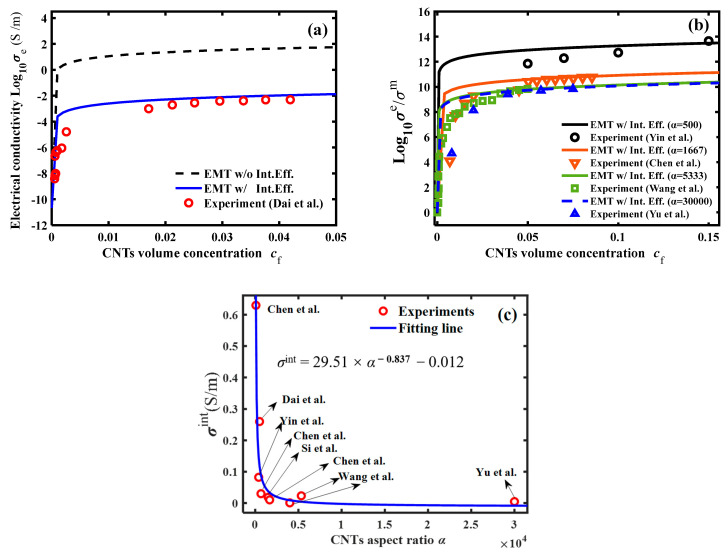
The electrical conductivity of the nanocomposite vs. the volume concentration of the CNTs nanofiller: (**a**) the comparison between the theoretical results and experimental measurements in literature [22,34,35,36,38,39,40,41]; (**b**) the electrical conductivity vs. the volume concentration of the CNTs nanofiller; and (**c**) the interfacial electrical conductivity of CNTs and epoxy vs. the aspect ratio of CNTs nanofiller.

**Table 1 polymers-16-00808-t001:** Physical values used in the calculation for comparison with Javadinejad et al. [44].

Material Parameters	Values
Average transverse size of CNTs, *l* (μm)	15
Mean diameter of CNTs, *λ* (nm)	7.5
Aspect ratio of CNTs, *α*	2000
Thickness of the interlayer, *δ* (nm)	0.75
Young’s modulus of epoxy, *Ε*^m^ (GPa)	2.6
Young’s modulus of CNTs, $E_{1}^{f}$, $E_{3}^{f}$ (GPa)	30, 1000
Young’s modulus of interfacial layer, *Ε*^int^ (GPa)	20

**Table 2 polymers-16-00808-t002:** The effect of the aspect ratio of CNT fillers on the interfacial Young’s modulus and overall Young’s modulus of CNTs-reinforced nanocomposites.

Aspect Ratio, α	Young’s Modulus of Interfacial Layer, *Ε*^int^ (MPa)	Young’s Modulus of Nanocomposites, *Ε*^e^ (*c*_f_ = 0.005) (GPa)	Reference
188	890	3.67	Giovannelli et al. [17]
500	320	3.61	Cha et al. [9]
571	210	3.58	Alishahi et al. [12]
583	280	3.46	Dehrooyeh et al. [18]
1167	220	2.91	Hsieh et al. [19]
2000	20	2.90	Javadinejad et al. [44]
5000	80	2.89	Mei et al. [20]
6000	60	2.89	Zarasvand et al. [21]
8333	40	2.88	Wang et al. [22]
10,000	22	2.83	

**Table 3 polymers-16-00808-t003:** Physical values used in the calculation for comparison with Yu et al. [36].

Material Parameters	Values
Average transverse size of CNTs, *l* (μm)	5
Mean diameter of CNTs, *λ* (nm)	20
Aspect ratio of CNTs, *α*	250
Thickness of the interlayer, *δ* (nm)	2
Thermal conductivity of epoxy, *κ*^m^ (W/mK)	0.223
Thermal conductivity of CNTs, $\kappa_{1}^{f}$, $\kappa_{3}^{f}$ (W/mK)	60, 3000
Thermal conductivity of the interfacial layer, *κ*^int^ (W/mK)	17

**Table 4 polymers-16-00808-t004:** The effect of the aspect ratio of CNT fillers on the interfacial thermal conductivity and overall thermal conductivity of CNTs-reinforced nanocomposites.

Aspect Ratio, α	Interface Thermal Conductivity, *κ*^int^ (W/mK)	Thermal Conductivity of Nanocomposites, *κ*^e^ (*c*_f_ = 0.08) (W/mK)	Reference
17	92	1.32	Wang et al. [33]
20	110	1.25	
100	24	0.99	Chen et al. [34,35]
200	25	0.89	
250	17	0.67	Yu et al. [36]
278	10	0.54	
300	8.8	0.50	Shi et al. [32]
400	12	0.45	Liang et al. [31]

**Table 5 polymers-16-00808-t005:** Physical values used in the calculation for comparison with Dai et al. [38].

Material Parameters	Values
Average transverse size of CNTs, *l* (μm)	0.6
Mean diameter of CNTs, *λ* (nm)	1.5
Aspect ratio of CNTs, *α*	400
Thickness of the interlayer, *δ* (nm)	0.15
Electrical conductivity of epoxy, *σ*^m^ (S/m)	2.11 × 10^−11^
Electrical conductivity of CNTs, $\sigma_{1}^{f}$, $\sigma_{3}^{f}$ (S/m)	5, 3000
Electrical conductivity of the interfacial layer, *σ*^int^ (S/m)	8.23 × 10^−2^

**Table 6 polymers-16-00808-t006:** The effect of the aspect ratio of CNT fillers on the interfacial electrical conductivity and overall electrical conductivity of CNT-reinforced nanocomposites.

Aspect Ratio, α	Interface Electrical Conductivity, *σ*^int^ (S/m)	Electrical Conductivity of Nanocomposites, *σ*^e^ (*c*_f_ = 0.05) (10^−3^ S/m)	Reference
80	630	22.9	Meguid et al. [39] Dai et al. [38]
400	82.3	13.2	
500	260	8.1	Yin et al. [41] Chen et al. [35]
667	30	0.44	
1500	18.6	0.72	Moisala et al. [40] Chen et al. [34]
1667	10.3	0.67	
4000	0.8	0.14	Wang et al. [22]

## Data Availability

Data are contained within the article.

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
