# Peer review of "Interfacial and Filler Size Effects on Mechanical/Thermal/Electrical Properties of CNTs-Reinforced Nanocomposites"

_polymers, 2024, doi:10.3390/polym16060808_

Round 1

Reviewer 1 Report

Comments and Suggestions for Authors

Thank you for very fruitful analytical work. The results obtained will undoubtedly be useful in the design of CNT-matrix systems. However, there are a number of issues requiring clarification.

1) Line 159 "Under the assumption of the same aspect ratio of CNTs-fillers and interfacial layers...". Please clarify where this assumption comes from and under what experimental conditions does this assumption become valid?

2) You refer to experimental work with MWCNTs. Does your model take this into account somehow? Is it possible to transfer this concept to single-walled CNTs?

3) To model the Young's modulus and elastic properties, parameters of CNTs and epoxy resin are used, similar to the source [44], with which the comparison is made. However, it is not obvious why such a value of "thickness of the interlayer" (10 times less than the nanotube diameter) was chosen for modeling. What caused choice "thickness of the interlayer"? Because it is selected diversely for different mechanical/thermal/electrical studies (even with respect to the size of the nanotube).

4)  In the final table for mechanical parameters, the numbers are not in italics, but in the rest - in italics. Please bring to a unified style.

Reviewer 2 Report

Comments and Suggestions for Authors

The presented paper shows the results of derivation and experimental verification of the united unified force-thermal-electrical modified effective medium theory that aims for the theoretical comprehensive prediction model of the overall Young's modulus, thermal conductivity, and electrical conductivity of the CNTs-reinforced nanocomposites. The novelty is a lack of a unified theoretical model to encompass all the above features. The paper is of publishable quality and fits the scope of the Polymer journal.

As the results are well presented and provide good insights for CNT-epoxy nanocomposites, the authors could provide some insights into further challenges/directions that could be summarised from the application of such theory. Perhaps some conclusions about the translation of such theory to other types of polymer matrices, etc. Readers would greatly benefit from such more generalized and far-reaching authors' insights. 

Comments on the Quality of English Language

minor proof-reading would beneficial

Reviewer 3 Report

Comments and Suggestions for Authors

This work presents a theoretical modeling of CNT-polymer nanocomposites to predict mechanical/thermal/electrical properties, which was compared to experimental values reported by others.   The theoretical prediction based on the interfacial interactions between polymer matrix and CNT could provide interesting aspects to the computational chemistry and materials science communities.   However, a few important pieces of information should be included/revised in the manuscript. 

In the introduction section, the authors stated “it is necessary to comprehensively consider the effects of various factors such as CNTs-polymer interfacial connection, CNTs-fillers aspect ratio, and dispersion state.”.  However, it seems that the authors mainly described only one factor (i.e., CNTs-fillers aspect ratio) in the results section.   It would be interesting if the authors could consider other factors. 

 As the degree of interfacial interactions greatly influences the physicochemical properties of the composite materials, the authors need to specify the type of interactions between CNT and polymer matrix (are these calculations based on hydrogen bonding and/or van der Waals attractive force?)  Thus, it would be much clearer if the authors could comment on the major interfacial interactions (via covalent, electrostatic, and/or van der Waals) between the polymer matrix and CNT that can control the overall properties, including mechanical/thermal/electrical properties.

Comments on the Quality of English Language

The overall writing seems to be clear. 
